# Heteroepitaxial vertical perovskite hot-electron transistors down to the monolayer limit

Brian S.Y. Kim[1,2]*, Yasuyuki Hikita[3], Takeaki Yajima[4] & Harold Y. Hwang[2,3]*

Two-dimensional heterostructures combined with vertical geometries are candidates to probe and utilize the physical properties of atomically-thin materials. The vertical configuration enables a unique form of hot-carrier spectroscopy as well as atomic-scale devices. Here, we present the room-temperature evolution of heteroepitaxial perovskite hot-electron transistors using a $SrRuO_3$ base down to the monolayer limit ($\sim 4\,\text{Å}$). As a fundamental electronic probe, we observe an abrupt transition in the hot-electron mean free path as a function of base thickness, coinciding with the thickness-dependent resistive transition. As a path towards devices, we demonstrate the integrated synthesis of perovskite one-dimensional electrical edge contacts using water-soluble and growth-compatible $Sr_3Al_2O_6$ hard masks. Edge-contacted monolayer-base transistors exhibit on/off ratios reaching $\sim 10^8$, complete electrostatic screening by the base manifesting pure hot-electron injection, and excellent scaling of the output current density with device dimensions. These results open new avenues for incorporating emergent phenomena at oxide interfaces and in heterostructures.

[1] Department of Electrical Engineering, Stanford University, Stanford, CA 94305, USA. [2] Geballe Laboratory for Advanced Materials, Department of Applied Physics, Stanford University, Stanford, CA 94305, USA. [3] Stanford Institute for Materials and Energy Sciences, SLAC National Accelerator Laboratory, Menlo Park, CA 94025, USA. [4] Department of Materials Engineering, The University of Tokyo, Bunkyo, Tokyo 113-8656, Japan. *email: bsk2137@columbia.edu; hyhwang@stanford.edu

Along with exfoliated two-dimensional (2D) materials[1–5], atomic-scale artificial perovskite oxide heterostructures offer exciting opportunities[6,7] for new devices and experimental probes of quantum materials. Specifically, designing vertical architectures incorporating these materials has unique advantages. It effectively utilizes their atomically thin nature by actively setting the channel length to an atomic scale, which would be an enabling pathway for fast electronics[8], which is technically challenging in planar devices. More importantly, the perpendicular geometry can diversify functionalities in fundamental studies and applications by integrating layers with a wide range of physical properties in an interface-specific manner. For example, hot-electron transport across interfaces can be used to surpass fundamental thresholds[9,10], such as the thermodynamic Shockley–Queisser limit in solar cells. In oxide heterostructures, hot-electron spectroscopy[11] would be an invaluable tool to investigate the nanometer scale electronic reconstructions that are often observed at interfaces and in confined geometries[7].

In pursuit of this approach, we demonstrate the evolution of a highly robust vertical hot-electron transistor (HET)[12] based on perovskite oxide heterostructures down to the monolayer-base limit, consisting of a $SrTiO_3$ emitter (both Nb-doped and undoped), atomically thin $SrRuO_3$ base, and $Nb:SrTiO_3$ (001) collector (Fig. 1; see the Methods section).

## Results

**Heterostructure fabrication and junction characteristics.** In device operation, hot electrons are injected out-of-plane across the forward-biased base–emitter (BE) junction, traverse the grounded base, and are collected across the reverse-biased base–collector (BC) junction. We first focus on devices with conventional laterally staggered surface contacts. The heteroepitaxial trilayer shows epitaxial and atomically flat surface topographies with the perovskite step-and-terrace structure of the underlying substrate, as seen by atomic force microscopy (AFM) and reflection high-energy electron diffraction (RHEED) patterns (Fig. 2a–e). Accordingly, both interfaces form Schottky junctions following the thermionic emission model, evident from the rectifying current-voltage characteristics with constant Schottky

barrier height $\phi$ and ideality factor $\eta$ near unity down to the monolayer limit (Fig. 2f, g). This is remarkable considering that only a single unit cell (u.c.) of $SrRuO_3$ (~4 Å) is responsible for the Schottky junction formation at both interfaces.

**Electrical characteristics of variable thickness $SrRuO_3$ HETs.** The common-base and common-emitter output characteristics of these devices show clear transconductance down to the monolayer limit (Fig. 3a–d). As the base becomes thinner, more hot electrons overcome $\phi_{BC}$ due to less scattering in the base; accordingly, the collector current density $J_C$ increases exponentially (Fig. 3b, inset). In general, an atomically thin base could be prone to the formation of pinhole and edge defects which electrostatically couple the emitter and the collector via a semiconducting channel, inducing permeable-base transistor operation[13,14] governed by the drift-diffusive transport of Fermi-level electrons, as opposed to hot-electron transport. To rule out this possibility, we demonstrate that the metal base in our HETs completely screens the electric field and electrically isolates the emitter and the collector, by the measurement of constant voltage feedback curves (Fig. 3a–c).

The common-base transfer characteristics further corroborate this complete screening in the base by showing that the $V_{EB}$ required for the onset of $J_C$ is independent of $V_{CB}$ (Fig. 3f). Considering the Thomas–Fermi screening length $L_{TF}$ of ~2 Å in $SrRuO_3$ at both interfaces, it is significant that the transistor still operates as a pure HET with only a single u.c. of $SrRuO_3$. The monolayer common-base current gain $\alpha \sim 0.12$ and common-emitter current gain $\beta \sim 0.14$ follow the expected relation $\beta = \alpha/(1-\alpha)$. After optimizing the processing conditions, our devices show 100% HET yield with exceptionally high stability and reproducibility for >100 devices examined (Fig. 3e). This is in contrast to prior work using a manganite base with low HET yield and dominantly permeable-base devices[14]. We attribute this important difference to the perfect registry of $SrRuO_3$ with the underlying substrate; namely, the common Sr cation throughout the trilayer, with interfaces free of polar electrostatic boundary conditions[15,16].

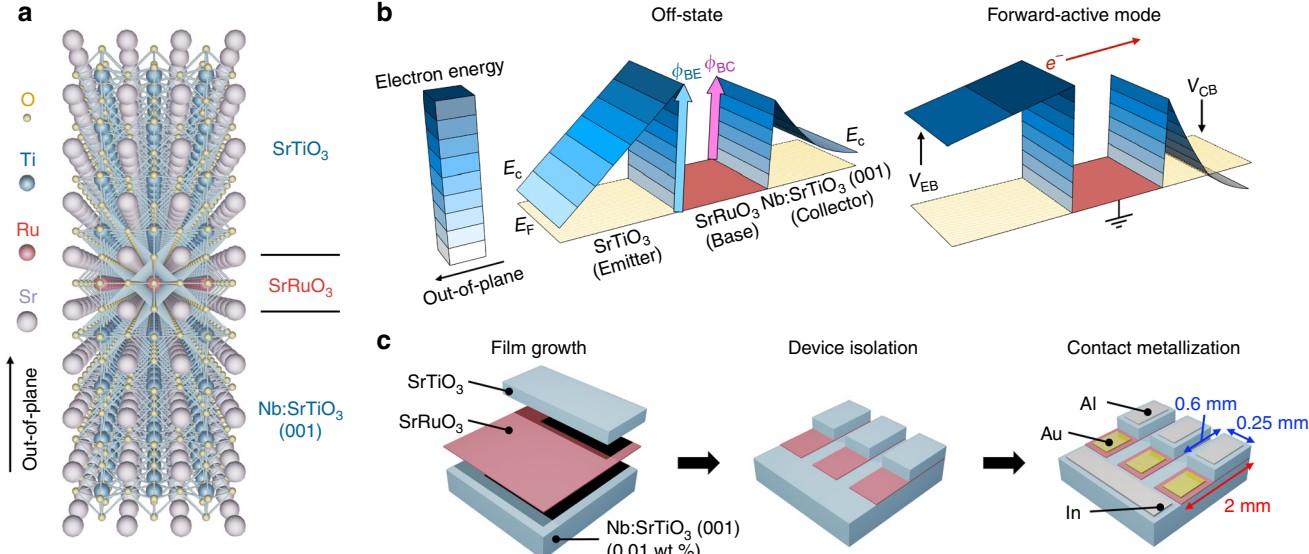

**Fig. 1** Schematics and working principle of the perovskite hot-electron transistor (HET). **a** The crystal structure of the atomically thin HET with a $SrTiO_3$ emitter/$SrRuO_3$ base/$Nb:SrTiO_3$ (001) collector trilayer. **b** Schematic illustration of the HET energy band diagram in the off-state (left) and in the common-base forward-active mode (right). In device operation, hot electrons are injected across the forward-biased base–emitter (BE) junction, traverse across the grounded base, and are collected across the reverse-biased base–collector (BC) junction. **c** Device process flow of the HET with laterally staggered electrical contacts.

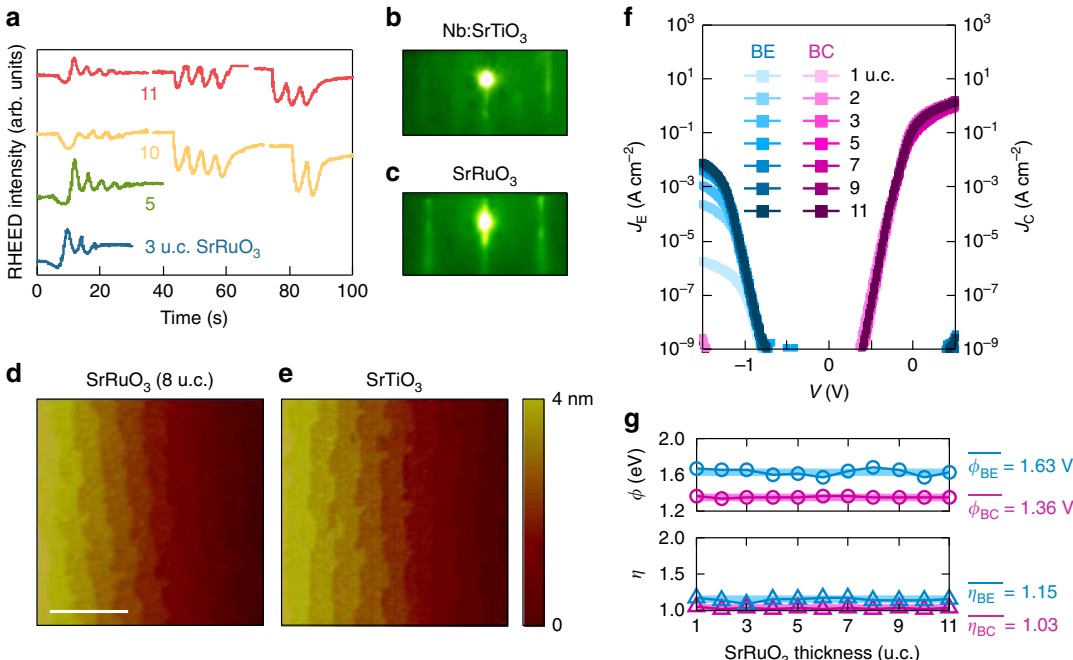

**Fig. 2** Heterostructure fabrication and room-temperature junction characteristics. **a** RHEED intensity oscillations for the growth of SrRuO$_3$ base. In situ RHEED patterns of **b** Nb:SrTiO$_3$ (001) and **c** SrRuO$_3$. AFM topographies of **d** 8 unit cell (u.c.) SrRuO$_3$, and **e** SrTiO$_3$ thin films, showing a clear perovskite step-and-terrace surface structure. The scale bar is 1 μm. **f** Current-voltage characteristics of the BC and BE Schottky junctions, for various SrRuO$_3$ thicknesses down to the monolayer limit. **g** Extracted BC and BE junction Schottky barrier height $\phi$ (top) and ideality factor $\eta$ (bottom). The solid lines are the mean values.

**Probing the fundamental characteristics of hot electrons**. The extremely high reliability of these HETs allows the experimental determination of hot-electron parameters, which were not previously accessible due to low yield and device-to-device fluctuations[14]. As shown in Fig. 4a–c, we observe the systematic thickness-dependent evolution of $\alpha$ over a wide range of ~7 orders of magnitude. Note that these devices were optimized for stability, not output current density, by incorporating a thick undoped SrTiO$_3$ emitter (Fig. 1). This avoided the growth-to-growth carrier density fluctuations of doped SrTiO$_3$, which would obscure the intrinsic response for thicker SrRuO$_3$. The exponential trend of $\alpha$ can be parametrized by the hot-electron mean free path $\lambda_H$ as[12]

$$\alpha = \begin{cases} \alpha_0 \times \exp(-t/\lambda_{H,I}), & t \leq 3 \text{ u.c.} \\ \alpha_0 \times \exp(-3 \text{ u.c.}/\lambda_{H,I}) \times \exp[-(t - 3 \text{ u.c.})/\lambda_{H,B}], & t > 3 \text{ u.c.} \end{cases}$$
(1)

where $t$ is the SrRuO$_3$ thickness, $\alpha_0$ is $\alpha$ extrapolated to $t = 0$ u.c., and $\lambda_{H,B}$ ($\lambda_{H,I}$) is the bulk (interfacial) hot-electron mean free path. The fact that $\alpha$ fits very well the exponential form self-consistently confirms HET operation, and highlights the ability to control SrRuO$_3$ thickness on the atomic level. These results indicate that SrRuO$_3$ is homogeneous with $\lambda_{H,I} \sim 0.3 \pm 0.01$ u.c. up to 3 u.c. However, two distinct regions exist for thicker SrRuO$_3$: the interfacial region with $\lambda_{H,I}$, and the 'bulk' region away from the interface with $\lambda_{H,B} \sim 1.4 \pm 0.03$ u.c., in good correspondence to recent studies using ballistic electron emission microscopy across the vacuum/SrRuO$_3$ interface[17].

The abrupt transition in $\lambda_H$ at 3 u.c. is quite reminiscent of the commonly observed resistivity transition for ultrathin SrRuO$_3$[18–21]. While there are some variations in the observed critical thickness, and debate on its origin, we see the same transition in our films. Figure 4c shows the thickness-dependent evolution of the room-temperature in-plane conductivity for identically grown samples on an undoped substrate. Notably, for 3 u.c. and below, the temperature dependence is insulating, while for 4 u.c. and above,

increasingly bulk-like metallic temperature dependence is observed (Fig. 4d, inset).

From a fundamental perspective, $\lambda_H$ has a rather interesting comparison to that for conventional transport, $\lambda_F$, the Fermi-electron mean free path. $\lambda_F$ can be estimated from the resistivity $\rho$, which can be expressed via Boltzmann transport as $\rho = 3\pi^2 \hbar / q^2 k_F^2 \lambda_F$, where $\hbar$ is the reduced Planck's constant, $q$ is the elementary charge, and $k_F$ is the Fermi wave vector[22]. Figure 4d shows the deduced thickness-dependent $\lambda_F$ at room temperature. As previously noted, the smooth decrease of $\lambda_F$ down to 4 u.c. arises from enhanced surface/interface scattering[20]. Even in the atomically clean limit, surface/interface scattering impacts the in-plane transport of ultrathin films[23], including diffuse scattering at step edges, whereby $\lambda_F$ can be expressed in terms of $t$ using Matthiessen's rule as $1/\lambda_F = 1/\lambda_{F,B} + 1/t_0 t$, where $\lambda_{F,B}$ is the bulk Fermi-electron mean free path, and $t_0$ is the surface/interface scattering constant. Estimating $\lambda_{F,B}$ as $\lambda_F$ for $t = 60$ u.c., $\lambda_F$ fits well this functional form down to 4 u.c. but then drastically decreases in the thinner films with insulating temperature dependence.

As has been previously emphasized, the extremely short $\lambda_{F,B}$ (300 K) ~ 0.6 u.c. in SrRuO$_3$ is highly inconsistent with coherent transport in conventional metals[22,24], and this 'badly metallic' behavior is a characteristic feature of strongly correlated metals under much current investigation[25]. Here we find that $\lambda_{H,B}$ exceeds our estimate for $\lambda_{F,B}$ by a factor of ~2.3 (Fig. 4d). This enhancement could originate from Ru 4$d$ electrons effectively screening the inelastic electron–electron scattering of hot electrons[26]. Note that $\lambda_{H,B}$ can be isolated from surface/interface scattering, as indicated by its constant thickness dependence above the sharp transition between 3 u.c. and 4 u.c.—this is a unique advantage of these thickness-dependent hot-electron experiments, enabling a direct measure of fundamental scattering mechanisms without surface/interface contributions. The surprising enhancement of $\lambda_{H,B}$ with respect to $\lambda_{F,B}$ suggests that further hot-electron spectroscopic studies of strongly correlated metals would provide valuable insights in this relatively unexplored regime.

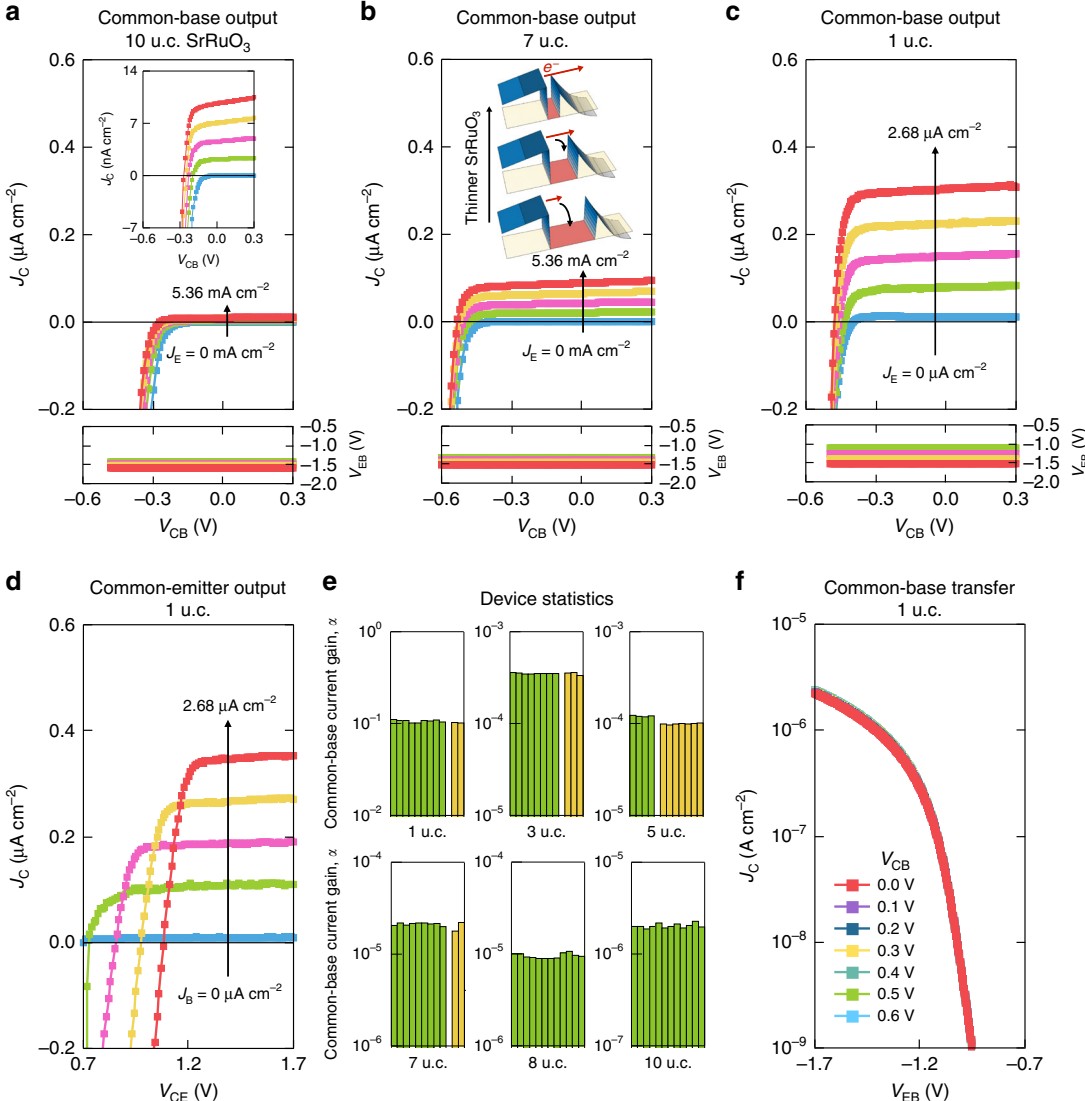

**Fig. 3** Room-temperature electrical characterization of variable thickness $SrRuO_3$ HETs. Common-base output characteristics (top) and voltage feedback (bottom) for $SrRuO_3$ thicknesses of **a** 10 u.c. and **b** 7 u.c., as $J_E$ is increased from 0 to 5.36 mA cm$^{-2}$, in 1.34 mA cm$^{-2}$ steps; and **c** 1 u.c., as $J_E$ is increased from 0 to 2.68 µA cm$^{-2}$, in 0.67 µA cm$^{-2}$ steps. The inset to **a** shows the region where clear transconductance can be seen for 10 u.c. $SrRuO_3$ thickness. The inset to **b** shows a schematic illustration of the increase of hot-electron transfer as the base becomes thinner. The constant voltage feedback curves in the common-base output characteristics demonstrate the complete electrostatic screening by the base metal. **d** Common-emitter output characteristics for 1 u.c. $SrRuO_3$ thickness, as $J_B$ is increased from 0 to 2.68 µA cm$^{-2}$, in 0.67 µA cm$^{-2}$ steps. **e** Device statistics on common-base current gain $\alpha$. For a specified $SrRuO_3$ thickness, each bar represents a single device and each color represents a distinct sample from a different fabrication run. **f** Common-base transfer characteristics for 1 u.c. $SrRuO_3$ thickness with $V_{CB}$ ranging from 0.0 to 0.6 V, in 0.1 V steps. $V_{EB}$ required for the onset of $J_C$ is independent of $V_{CB}$, indicating pure HET operation.

**Perovskite one-dimensional electrical edge contacts**. We turn now to considerations relevant for the potential development of monolayer heterostructures for practical devices. While there are several important issues that need to be addressed, a central obstacle is the low output current density in the HETs presented thus far ($J_C \sim$ µA cm$^{-2}$). Two significant contributions are the high emitter resistance introduced via thick undoped $SrTiO_3$ (needed for device stability investigating the thick-base regime, but which limits hot-electron injection), and the in-plane base series resistance arising from the laterally staggered contact geometry, which becomes increasingly dominant in the monolayer limit. In order to address these issues, we can thin down and dope the emitter, which readily improves the emitter resistance by orders of magnitude[15,27,28]. Furthermore, we demonstrate here the synthesis of a perovskite one-dimensional electrical edge

contact, in analogy to that recently developed for 2D materials[29], using a water-soluble and growth-compatible $Sr_3Al_2O_6$ hard mask layer[30] (Fig. 5a; see the Methods section). The $Sr_3Al_2O_6$ layer not only bypasses aggressive lift-off processes, but also is stable at high growth temperatures and oxidizing conditions, allowing for the corresponding growth of the perovskite edge contact.

As a result, the monolayer-base devices improved significantly: $J_C$ increased by ~5 orders of magnitude; on/off current ratios reached ~100,000,000; and $\alpha$ increased threefold to ~0.35, with a corresponding enhancement of $\beta$ to ~0.54 (Fig. 5). Pure HET operation is evident from the voltage feedback curves and the transfer characteristics as discussed previously (Fig. 5b, d). Most importantly, we find that $J_C$ self-consistently scales with the residual base resistance underneath the emitter as $1/R_E$ ($R_E$ is the emitter radius)[31], which is very favorable to device downscaling

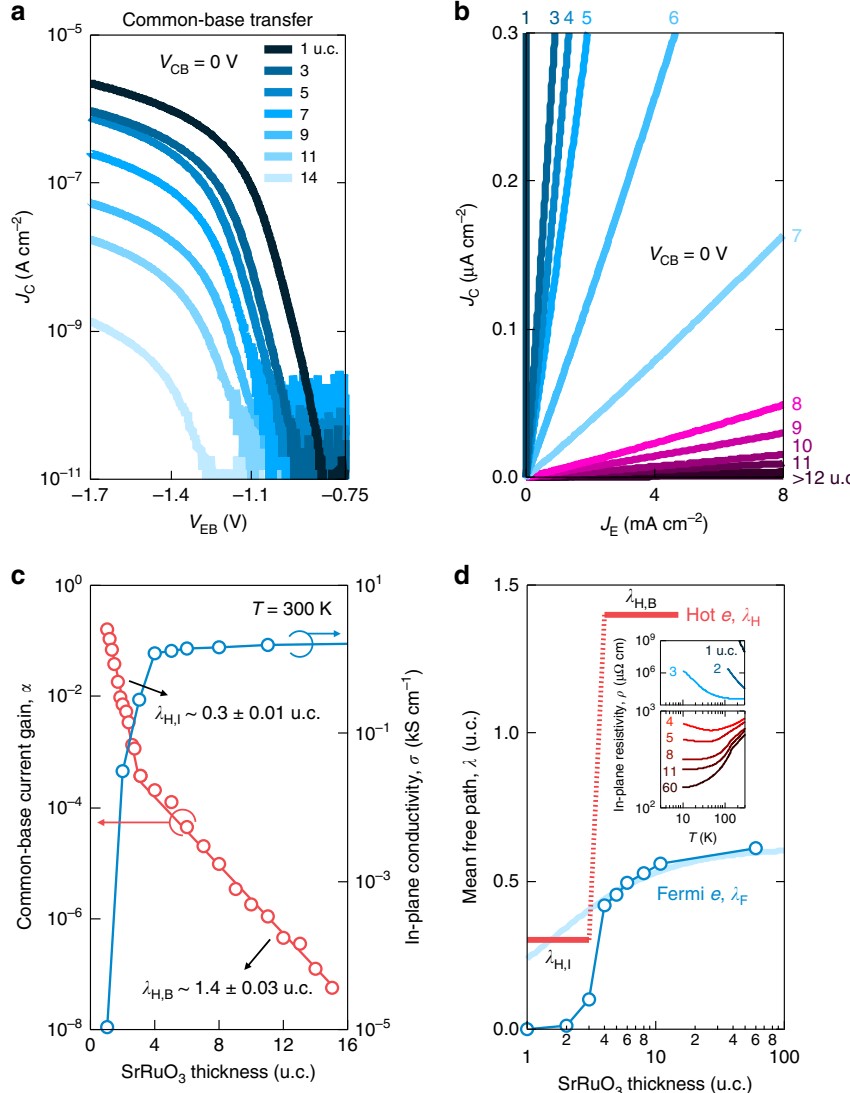

**Fig. 4** Probing the fundamental characteristics of hot electrons down to the monolayer limit of the HET at room temperature. Common-base transfer characteristics for different SrRuO₃ thicknesses at $V_{CB} = 0$ V, plotted as **a** $V_{EB}$ versus $J_C$, and **b** $J_E$ versus $J_C$. **c** Common-base current gain $\alpha$ (red open circles) at $V_{CB} = 0$ V and in-plane conductivity $\sigma$ of the SrRuO₃/SrTiO₃ (001) heterostructures (blue open circles) as a function of SrRuO₃ thickness at room temperature. $\alpha$ spans a wide range of ~7 orders of magnitude and self-consistently follows an exponential trend, illustrating the HET operation of the devices. Two distinct exponential regimes exist as a function of SrRuO₃ thickness, where the deduced hot-electron mean free path $\lambda_H$ is ~0.3 ± 0.01 u.c. from the data up to 3 u.c., and ~1.4 ± 0.03 u.c. from the data for thicker base. Below 3 u.c., fractional coverage devices smoothly interpolate. The red solid lines are best linear fits to the two regimes. **d** $\lambda_H$ near the center of the SrRuO₃ base layer (red solid line) and Fermi-electron mean free path $\lambda_F$ (blue open circles) as a function of SrRuO₃ thickness. The quasi-universal thickness-dependent metal–insulator transition is evident both from $\lambda_H$ and $\lambda_F$ across the same critical thickness. The blue solid line is a fit assuming a dominant surface/interface scattering (see text). The inset shows the thickness-dependent in-plane resistivity $\rho$ as a function of temperature $T$ for SrRuO₃/SrTiO₃ (001) heterostructures.

(Fig. 5g). This confirms that the perovskite edge contact effectively minimizes the base series resistance by placing the contact at the edge of the monolayer base/emitter interface, and should enable high current density in sub-micron devices. Furthermore, the edge contact method developed here should be broadly useful for a wide range of oxide heterostructure devices.

## Methods
**Heterostructure fabrication and device processing**. The heteroepitaxial trilayer was fabricated using pulsed laser deposition with a 248 nm KrF excimer laser using TiO₂-terminated Nb:SrTiO₃ (001) substrates (0.01 wt% Nb dopant concentration). SrRuO₃ base (1 u.c. to 15 u.c.) was grown in 100 mTorr O₂. The base thickness was controlled on the atomic scale by monitoring the in situ RHEED oscillations, where

one oscillation corresponds to a single u.c. Coherent RHEED oscillations throughout the growth indicate high quality layer-by-layer growth of the SrRuO₃. In particular, the growth of SrRuO₃ is interrupted every 4 u.c. for thicker films so that the adatoms on the surface can sufficiently rearrange, and preserve the step-and-terrace structure of the underlying substrate, as indicated by the RHEED intensity recovery right after the interruption (Fig. 2a). This preservation of the step-and-terrace structure is especially critical for SrRuO₃, because it has been otherwise reported to adversely affect the transport properties[20]. A ~60 nm-thick SrTiO₃ emitter was consecutively grown through a rectangular template mask in 1 mTorr O₂. The substrate temperature was 700 °C during the entire growth. The SrRuO₃/SrTiO₃ (001) heterostructure was fabricated under the same growth conditions. The SrTiO₃/SrRuO₃/Nb:SrTiO₃ (001) trilayer was then Ar-ion etched into rectangular shapes, and annealed in 760 Torr O₂ at 350 °C for 6 h to fill oxygen vacancies generated by Ar-ion etching. Rectangular Au and Al electrodes were deposited using e-beam evaporation on the base and emitter, respectively. Indium was ultrasonically soldered onto the collector. These electrodes form Ohmic contacts to each layer.

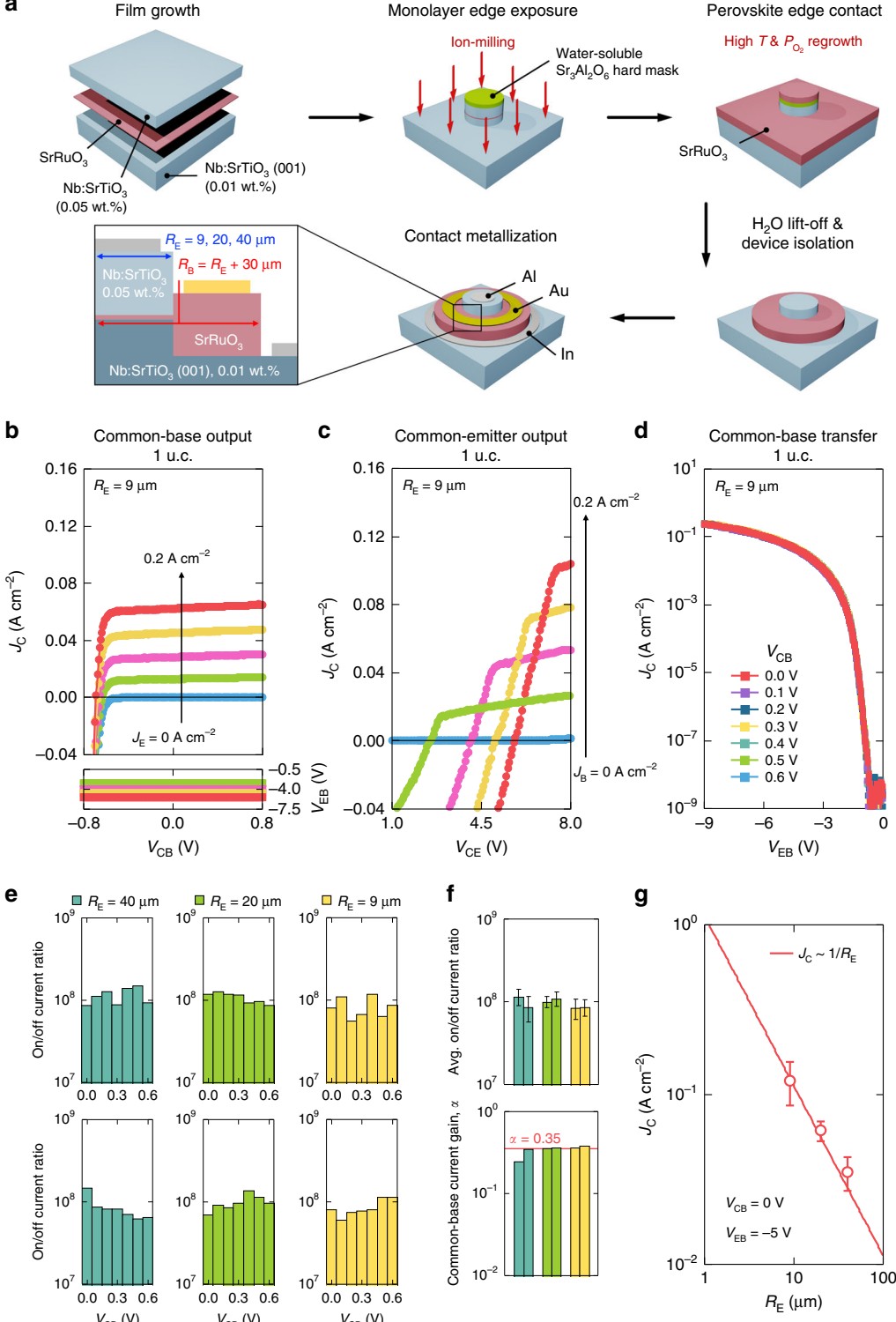

**Fig. 5** Room-temperature electrical characterization of monolayer-base HETs with a perovskite one-dimensional electrical edge contact. **a** Schematic of the perovskite edge contact fabrication process using a water-soluble and growth-compatible $Sr_3Al_2O_6$ hard mask layer[30]. $R_E$ ($R_B$) is the emitter (base) radius. **b** Common-base output characteristics (top) and voltage feedback (bottom), as $J_E$ is increased from 0 to 0.2 A cm$^{-2}$, in 0.05 A cm$^{-2}$ steps. **c** Common-emitter output characteristics, as $J_B$ is increased from 0 to 0.2 A cm$^{-2}$, in 0.05 A cm$^{-2}$ steps. **d** Common-base transfer characteristics with $V_{CB}$ ranging from 0.0 to 0.6 V, in 0.1 V steps. $R_E$ is 9 μm for the devices shown in **b** through **d**. **e** On/off current ratio as a function of $V_{CB}$ ranging from 0.0 to 0.6 V, in 0.1 V steps. Each panel represents a different device. **f** On/off current ratio averaged over $V_{CB}$ ranging from 0.0 to 0.6 V (top) and common-base current gain $\alpha$ (bottom) for devices shown in **e**. The red solid line represents $\alpha = 0.35$, each bar represents a different device, and the error bars represent the standard deviations. The blue, green, and yellow solid bars in **e** and **f** represent devices with $R_E = 40$, 20, and 9 μm, respectively. **g** $J_C$ as a function of $R_E$ at $V_{CB} = 0$ V and $V_{EB} = -5$ V. The error bars represent the standard deviations. The data (red open circles) self-consistently scale with the residual base resistance underneath the emitter as $J_C \sim 1/R_E$ (red solid line)[31].

**Perovskite one-dimensional electrical edge contact**. ~30 nm-thick Nb:SrTiO$_3$ (0.05 wt% Nb dopant concentration)/monolayer SrRuO$_3$/Nb:SrTiO$_3$ (001) heteroepitaxial trilayer was grown using the same substrates and growth conditions as noted above (the emitter was grown without the use of a template mask). The key step is the growth of a water-soluble and growth-compatible Sr$_3$Al$_2$O$_6$ hard mask layer on the trilayer. The Sr$_3$Al$_2$O$_6$ hard mask layer bypasses aggressive lift-off processes and also serves as a mask during Ar-ion etching and perovskite edge contact regrowth processes involving high growth temperatures and oxidizing conditions. A ~180 nm-thick Sr$_3$Al$_2$O$_6$ hard mask layer[30] was grown in lithographically patterned circular shapes in $5 \times 10^{-6}$ Torr O$_2$ at room temperature, and capped with a ~60 nm-thick SrTiO$_3$ grown in 1 mTorr O$_2$ at room temperature to enhance its stability in the air. The trilayer covered with the Sr$_3$Al$_2$O$_6$ hard mask layer in circular shapes was Ar-ion etched to expose the edge of the monolayer base and annealed in 760 Torr O$_2$ at 350 °C for 6 h. A ~30 nm-thick SrRuO$_3$ edge contact was consecutively grown in 100 mTorr O$_2$ at 700 °C. The entire structure was then immersed into room-temperature filtered de-ionized water to dissolve the Sr$_3$Al$_2$O$_6$ hard mask layer. For device isolation, the trilayer was Ar-ion etched into larger circular shapes and annealed in 760 Torr O$_2$ at 350 °C for 6 h. Electrodes were deposited on each layer as noted above.

**Characterization**. AFM images were acquired in tapping mode. All electrical measurements of the device were performed using a semiconductor parameter analyzer in DC mode at room temperature in ambient conditions. The in-plane transport measurements were conducted in a four-point geometry with Au contacts for SrRuO$_3$ films.

## Data availability
The data that support the findings of this study are available from the corresponding authors upon reasonable request.

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

## Acknowledgements
This work was supported by the Department of Energy, Office of Basic Energy Sciences, Division of Materials Sciences and Engineering, under Contract no. DE-AC02-76SF00515. Work on the one-dimensional electrical edge contact was supported by the Gordon and Betty Moore Foundation's EPiQS Initiative through Grant GBMF4415.

## Author contributions
B.S.Y.K. performed the device fabrication, measurements, and data analysis. Y.H., T.Y., and H.Y.H. assisted with the planning, measurements, and analysis of the study. B.S.Y.K. wrote the manuscript, with input from all authors.

## Competing interests
The authors declare no competing interests.
