## [Peer Review File · Nature Communications]

Reviewers Comments:

Reviewer #1 (Remarks to the Author):

In this manuscript, the authors report on the DC characteristics of a vertical hot-electron transistor based on perovskite oxide heterostructures. The reported devices consist of SrTiO₃ as the emitter/collector and SrRuO₃ (down to one unit cell thick) as the base material. The authors claim achieving six orders of magnitude on/off current ratio and a subthreshold swing (SS) reaching 15.5 mV/dec. Furthermore, hot-electron mean free path and thickness-dependent evolution of the resistivity in SrRuO₃ is studied and discussed. In general, this is an interesting work giving insights into fundamentals and characteristics of oxide heterostructures thanks to the optimized process and careful characterization. On the other hand, from the device performance perspective, it is far from being a competitive device or envisioned for any electronic application (digital, analog, high frequency, sensing, etc.) except for fundamental studies. The following explains, in more details, the reviewer's main concerns about the reported devices:

- One of the main claims of the paper is the subthermionic SS of 15.5 mV/dec. Considering the injection Schottky diode and the biasing condition, the achieved SS is contradicting the thermionic limit theory. This might be originating from a problem in the structure, i.e. a resistor network, or measurement unless the authors explain otherwise. If there is no such an explanation, this subthermionic SS cannot be introduced as one of the main claims of the paper. In addition, the achieved SS is in the collector current range of several tens of fA. Conclusions from measurements in such low current densities are not convincing.

- Another major concern arises over the extreme low current levels in these devices. The numbers are provided as absolute numbers, not normalized for device area. This makes an assessment of the results difficult. The device's dimensions are not presented, however, from the device structure and the schematic in figure one, the reviewer assumes an emitter area of about the same as the probing contact pad. This means that the on-current is smaller than or in the range of pA/ μm^2 . Consequently, substantial improvement, in the collector on-current density, is needed before considering it for any electronic device application. The reviewer notes that the doping of the emitter can improve the emitter current. However, this does not necessarily result in the same improvement in the Collector current. In other words, Higher doping may introduce other contributors to the emitter current through tunneling and traps which can affect the common-base current gain, SS, and the collector on-current. Furthermore, even the current level shown in Ref 16 is not high enough to be competitive with the state-of-the-art devices for high speed/high frequency applications.

- While the devices, even with the monolayer base, show surprisingly decent electric field screening, the common-base current gain is very poor. Especially for More than Moore applications, the current gain is a very critical figure of merit. Unfortunately, the schemes proposed, by the authors, to enhance the current gain are not concrete and convincing.

- Looking at the I-V characteristics of the BE Schottky junction in figure 2f as well as the in-plane resistivity shown in figure 5d (inset), the devices with 1 u.c. and even 2 u.c. of SrRuO₃

introduce a large base and series resistance to the device. Even from 2 u.c. to 1 u.c. thick base, there is about two orders of magnitude increase in the base series resistance. This is detrimental for device high speed performance. One should note that the potential high-speed/high frequency performance is one of the main advantages of vertical hot-electron transistors.

- The six orders of magnitude on/off current ratio, which is one of the main claims of the paper should be explicitly shown in the paper.

Considering the above comments and the fact that a vertical hot-electron transistor based on perovskite materials has been previously reported (Yajima et al., Nature Mater, 2011), the reviewer does not recommend a potential publication of this manuscript in Nature Nanotechnology.

Reviewer #2 (Remarks to the Author):

In this manuscript, Kim and collaborators report on the fabrication and characterization of heteroepitaxial perovskite vertical hot-electron transistors based on SrRuO₃ monolayer base. They discuss the room-temperature operation and the evolution of device properties as a function of layer thickness. These results are important towards the scaling down of electronic devices. The article is interesting and it is well written. I recommend publication after addressing the following comments:

1. The use of term “robust” and “extremely high stability and reproducibility” when describing the devices developed here needs more justification. Do the authors refer to the operational and/or environmental stability? If yes, more details should be provided. On the other hand, if they mean that the fabrication method is robust, since they claim that they have 100 % yield, they should provide more details on that. At the moment, on page 4 they simply mention that “after optimizing the processing conditions” the yield became 100%. This is true both for the monolayer and for the multilayer transistors, but in particular for the latter.

2. How uniform is the performance of the devices obtained on the same substrate? How about the differences between the different fabrication runs. More statistics should be provided for the device properties.

3. A serious challenge in miniaturizing field-effect transistors is the fact that the contribution from contacts becomes dominant at short channel lengths. How important is the quality of contacts in this case? A discussion on this topic would strengthen the manuscript, as well as the motivation that the authors proposed.

4. The authors should include a graph of the dependence of the on/off ration on the applied voltage, and they should also provide more details about the contact geometry and size.

Reviewer #1 (Remarks to the Author):

Overall Comment: In this manuscript, the authors report on the DC characteristics of a vertical hot-electron transistor based on perovskite oxide heterostructures. The reported devices consist of SrTiO₃ as the emitter/collector and SrRuO₃ (down to one unit cell thick) as the base material. The authors claim achieving six orders of magnitude on/off current ratio and a subthreshold swing (SS) reaching 15.5 mV/dec. Furthermore, hot-electron mean free path and thickness-dependent evolution of the resistivity in SrRuO₃ is studied and discussed. In general, this is an interesting work giving insights into fundamentals and characteristics of oxide heterostructures thanks to the optimized process and careful characterization. On the other hand, from the device performance perspective, it is far from being a competitive device or envisioned for any electronic application (digital, analog, high frequency, sensing, etc.) except for fundamental studies. The following explains, in more details, the reviewer's main concerns about the reported devices.

Overall Response: We thank the reviewer for their time and comments. In responding to their critique, we first note that the reviewer solely focuses on the device aspects for applications, while we believe the importance and relevance for this journal are equally on the fundamentals of nanoscale oxide heterostructures. Nevertheless, we followed the reviewer's emphasis in our response and revisions. Following this emphasis, we have removed the presentation and discussion of the subthermionic SS; an expanded presentation of this aspect of the work does not provide a direct pathway to high current densities, which appears to be the priority of the reviewer. To that end, we have performed a serious expansion of the work, both in terms of devices with doped emitters, as well as developing and incorporating one-dimensional electrical edge contacts. We hope the reviewer will agree that we have taken significant steps to address the priorities conveyed in their review. Please find our detailed response to the reviewer's comments below. The line and figure numbers mentioned in our response correspond to those in the revised manuscript.

Comment 1: One of the main claims of the paper is the subthermionic SS of 15.5 mV/dec. Considering the injection Schottky diode and the biasing condition, the achieved SS is contradicting the thermionic limit theory. This might be originating from a problem in the structure, i.e. a resistor network, or measurement unless the authors explain otherwise. If there is no such an explanation, this subthermionic SS cannot be introduced as one of the main claims of the paper. In addition, the achieved SS is in the collector current range of several tens of fA. Conclusions from measurements in such low current densities are not convincing.

Response 1: As noted above, we have removed this aspect of the paper. As a next series of experiments, we intend to study the low-temperature and magnetic-field-dependent response of these device structures. This will provide a more natural context in which to present and discuss the subthermionic SS. We note that the work by Sarkar *et al.*¹, as well as related and subsequent work, has generated considerable interest and controversy – the former in terms of potential low-energy devices, and the latter in part due to the low current densities involved. In this regime, we believe our devices likely exhibit related transport mechanisms, for which temperature dependence is helpful to elucidate.

However, this does not provide a directly apparent path for high current densities, which we have taken as the main focus of our revisions.

References

1. Sarkar, D. *et al.* A subthermionic tunnel field-effect transistor with an atomically thin channel. *Nature* **526**, 91–95 (2015).

Comment 2: Another major concern arises over the extreme low current levels in these devices. The numbers are provided as absolute numbers, not normalized for device area. This makes an assessment of the results difficult. The device's dimensions are not presented, however, from the device structure and the schematic in figure one, the reviewer assumes an emitter area of about the same as the probing contact pad. This means that the on-current is smaller than or in the range of pA/ μm^2 . Consequently, substantial improvement, in the collector on-current density, is needed before considering it for any electronic device application. The reviewer notes that the doping of the emitter can improve the emitter current. However, this does not necessarily result in the same improvement in the Collector current. In other words, Higher doping may introduce other contributors to the emitter current through tunneling and traps which can affect the common-base current gain, SS, and the collector on-current. Furthermore, even the current level shown in Ref 16 is not high enough to be competitive with the state-of-the-art devices for high speed/high frequency applications.

Response 2: We have revised the manuscript to include the device dimensions in Fig. 1c and Fig. 5a, and replaced all current values with current density, in units of A/cm². We have performed new experiments and added a demonstration of enhancing the collector current levels in lines 132–157 and Fig. 5 (please refer to the Methods section for detailed information on the device processing), which exactly follows our description in the original manuscript. Namely, the first limiting step is the non-doped emitter structure; the second is the base series resistance. We have addressed the former by doping and thinning the emitter, and the latter by developing and adopting a one-dimensional electrical edge contact method. We are quite excited by these results, which incorporate a novel processing and epitaxial edge growth scheme, which should be valuable for many situations where edge contacts to buried ultrathin oxide layers are required.

These modifications to the device structure allowed us to substantially improve the collector current density by ~5 orders of magnitude from $\sim 2.2 \times 10^{-6}$ A/cm² to $\sim 2.4 \times 10^{-1}$ A/cm² for relatively large devices. Furthermore, the common-base current gain increased by threefold from ~ 0.12 to ~ 0.35 . These results strongly indicate that for our devices, higher doping of emitters does not introduce major contributors that adversely affect the device performance (at room temperature). Therefore, further optimization of the emitter structure and the base contact geometry based on the methodology reported in our revised manuscript, together with the downscaling of the devices using nanolithography techniques, could readily enhance the collector current density much further, following the scaling trends of Fig. 5g. We also note that the current level shown in the report by Hikita *et al.*¹ does not represent the intrinsic upper limit for devices based on SrRuO₃. The devices reported by Hikita *et al.* were not specifically optimized for practical devices, and were rather load- or series-resistance limited, which could be

readily overcome by the edge contact method presented in our revised manuscript. It is also worth noting that the current level of our devices in Fig. 2f is already more than one order of magnitude higher than that of the devices reported by Hikita *et al.*

References

1. Hikita, Y., Kozuka, Y., Susaki, T., Takagi, H. & Hwang, H. Y. Characterization of the Schottky barrier in SrRuO₃/Nb:SrTiO₃ junctions. *Appl. Phys. Lett.* **90**, 143507 (2007).

Comment 3: While the devices, even with the monolayer base, show surprisingly decent electric field screening, the common-base current gain is very poor. Especially for More than Moore applications, the current gain is a very critical figure of merit. Unfortunately, the schemes proposed, by the authors, to enhance the current gain are not concrete and convincing.

Response 3: We certainly agree that current gain is important, and we note that the current gain (~0.35) as a result of the additional experiments is already relevant for high-speed applications. Given that the reviewer finds our proposals unconvincing, we have revised the manuscript to incorporate the improvements we have demonstrated, and removed other discussion. In terms of DC response, we believe our work represents the record gain in demonstrated hot-electron transistor (HET) operation (see Fig. R1 below). We note that there are many recent papers using exfoliated materials (e.g., graphene, transition metal dichalcogenides) that either explicitly or implicitly claim HET response. However, in all papers of which we are aware, the device characteristics are either clearly incompatible with HET operation (showing for example incomplete base screening), or at best inconclusive in one case. Given these circumstances, we have chosen not to emphasize this record value to avoid a contentious debate and discussion of the recent literature and rather let these results speak for themselves.

Shown below are the highest reported values of DC common-base current gain for devices where HET response has been experimentally established and fully checked. We also include data for the prior oxide HET report for later discussion.

Figure R1 | DC common-base current gain for devices where hot-electron transistor (HET) response has been experimentally established and fully checked. The data for La_{0.7}Sr_{0.3}MnO₃ (CoSi₂) was adapted from Yajima *et al.*¹ (Rosencher *et al.*^{2,3}).

References

1. Yajima, T., Hikita, Y. & Hwang, H. Y. A heteroepitaxial perovskite metal-base transistor. *Nature Mater.* **10**, 198–201 (2011).
2. Rosencher, E. *et al.* Realization and electrical properties of a monolithic metal-base transistor: The Si/CoSi₂/Si structure. *Physica B* **134**, 106 (1985).
3. Rosencher, E. *et al.* Study of ballistic transport in Si-CoSi₂-Si metal base transistors. *Appl. Phys. Lett.* **49**, 271 (1986).

Comment 4: Looking at the I-V characteristics of the BE Schottky junction in figure 2f as well as the in-plane resistivity shown in figure 5d (inset), the devices with 1 u.c. and even 2 u.c. of SrRuO₃ introduce a large base and series resistance to the device. Even from 2 u.c. to 1 u.c. thick base, there is about two orders of magnitude increase in the base series resistance. This is detrimental for device high speed performance. One should note that the potential high-speed/high frequency performance is one of the main advantages of vertical hot-electron transistors.

Response 4: As discussed in Response 2, we have addressed this using one-dimensional electrical edge contacts that we have developed for this purpose.

Comment 5: The six orders of magnitude on/off current ratio, which is one of the main claims of the paper should be explicitly shown in the paper.

Response 5: We now explicitly show the on/off current ratio reaching ~8 orders of magnitude in Fig. 5d–f. This is two orders of magnitude higher than that in the original manuscript due to the modifications to the device structure and a subsequent increase of the collector current density as described in Response 2.

Comment 6: Considering the above comments and the fact that a vertical hot-electron transistor based on perovskite materials has been previously reported (Yajima *et al.*, *Nature Mater.*, 2011), the reviewer does not recommend a potential publication of this manuscript in *Nature Nanotechnology*.

Response 6: With respect, we strongly disagree with the assessment that the earlier report on a perovskite oxide hot-electron transistor by Yajima *et al.*¹ (which was published by our group) undermines the significance of our current work. Eight years after its publication, this remains the only such experimental demonstration. Despite this, it has been cited ~115 times, and has had broad impact in related attempts in other two-dimensional materials, as well as the oxide heterostructure community. Ever since this first study, our group has worked to make fundamental advances that would enable the broad implementation and impact of this approach. This has included exploration of a range of materials for the base. After much time and effort, here we report such a breakthrough, with over 3 orders of magnitude increased current gain (see Fig. R1 above).

The earlier report, which focused solely on a very thick 50 unit cell thick base, required the atomic-scale synthesis of a complex interfacial structure (to adjust the band alignment) for proper device operation and suffered from substantial dispersion of device characteristics due to imperfect electric field screening in the base. That is to say (and as we had clearly reported), the vast majority of the devices operated in the permeable-base regime, and not as an HET. These imperfections fundamentally limited the (vertical) downscaling of those HET attempts since they are universally encountered as the base is thinned down. Accordingly, the device yield of the earlier report asymptotically approached a very low value of ~6%, even for the thick base used.

Therefore, our current work is a significant breakthrough overcoming the abovementioned limitations using a different base metal with new and precise control over device processing to achieve 100% yield with extremely high stability. We have further shown that our devices scale favorably down to the monolayer limit, i.e., 1/50th of the prior demonstration. While we would certainly agree that this should not be the final word on oxide devices in this geometry, it is in our opinion a major advance. In particular, it is much more straightforward to synthesize these structures, and as such we anticipate it will be much more accessible to the community. Despite the final assessment of the reviewer, we do not agree with the implication that our current work is a simple incremental extension of the earlier report.

References

1. Yajima, T., Hikita, Y. & Hwang, H. Y. A heteroepitaxial perovskite metal-base transistor. *Nature Mater.* **10**, 198–201 (2011).

Reviewer #2 (Remarks to the Author):

Overall Comment: In this manuscript, Kim and collaborators report on the fabrication and characterization of heteroepitaxial perovskite vertical hot-electron transistors based on SrRuO₃ monolayer base. They discuss the room-temperature operation and the evolution of device properties as a function of layer thickness. These results are important towards the scaling down of electronic devices. The article is interesting and it is well written. I recommend publication after addressing the following comments:

Overall Response: We thank the reviewer for the positive review of our work and the recommendation to publish. Please find our detailed response to the reviewer's comments below. The line and figure numbers mentioned in our response correspond to those in the revised manuscript.

Comment 1: The use of term “robust” and “extremely high stability and reproducibility” when describing the devices developed here needs more justification. Do the authors refer to the operational and/or environmental stability? If yes, more details should be provided. On the other hand, if they mean that the fabrication method is robust, since they claim that they have 100 % yield, they should provide more details on that. At the moment, on page 4 they simply mention that “after optimizing the processing conditions” the yield became 100%. This is true both for the monolayer and for the multilayer transistors, but in particular for the latter.

Response 1: We agree this was unclear. By the use of the terms “robust” and “extremely high stability and reproducibility”, we meant to describe the device yield, as evidenced by uniformity and systematic tunability of the device characteristics as the base layer thickness is varied on the atomic scale. In revised form, experimental details and statistics quantifying the uniformity can be found in Fig. 3e, as discussed in Response 2. The systematic tunability of our devices is evidenced from the fact that their thickness-dependent current gain spans a strikingly broad range of ~7 orders of magnitude and fits well the theoretical exponential functional form as we tune the base thickness in an atomically precise manner (please refer to lines 84–85 and Fig. 4c).

Comment 2: How uniform is the performance of the devices obtained on the same substrate? How about the differences between the different fabrication runs. More statistics should be provided for the device properties.

Response 2: We have included device statistics for both the devices obtained on the same substrate and between different fabrication runs in Fig. 3e. We have shown the common-base current gain of 68 devices for the base layer thicknesses of 1 unit cell (u.c.), 3 u.c., 5 u.c., 7 u.c., 8 u.c., and 10 u.c. The mean coefficient of variation (i.e., the ratio of the standard deviation to the mean) for these samples is 3.08%.

Comment 3: A serious challenge in miniaturizing field-effect transistors is the fact that the contribution from contacts becomes dominant at short channel lengths. How important is the quality of contacts in this case? A discussion on this topic would strengthen the manuscript, as well as the motivation that the authors proposed.

Response 3: We agree with the reviewer that a discussion of contacts and their scaling would strengthen the manuscript. As field-effect transistors (FET) scale down, the contribution from contact resistance becomes dominant not only due to reduced channel resistance, but also due to simultaneously reduced source/drain junction depth, i.e., “shallow” junction, which is essential to avoid short-channel effects such as threshold voltage roll-off. Analogously, contact resistance could become one of the major contributors in the resistor network of hot-electron transistors as they scale down. Given the reviewer’s interest, and the critique of related issues by reviewer #1, we spent considerable effort in the revised manuscript to develop and demonstrate one-dimensional electrical edge contacts, which provide excellent contacts to the monolayer base. Furthermore, we find that the device scaling characteristics are highly favorable (please refer to lines 132–157 and Fig. 5).

Comment 4: The authors should include a graph of the dependence of the on/off ration on the applied voltage, and they should also provide more details about the contact geometry and size.

Response 4: We have included the dependence of the on/off current ratio as a function of V_{CB} for multiple devices in Fig. 5d–f, and the contact geometry and size in Fig. 1c and Fig. 5a.

Reviewer #1 (Remarks to the Author):

Overall Comment: The authors have addressed the concerns raised by both reviewers and significantly improved the paper. I believe this paper, in its current form, is suitable for publication in *Nature Communications*.

Overall Response: We thank the reviewer for their time and recommendation for publication of our manuscript in *Nature Communications*.

Reviewer #2 (Remarks to the Author):

Overall Comment: The authors have addressed my comments adequately and I recommend publication in current form.

Overall Response: We thank the reviewer for their time and recommendation for publication of our manuscript in *Nature Communications*.